**The Rocklea Dome 3D Mineral Mapping Test Data Set**
Carsten Laukamp[a], Maarten Haest[b], Thomas Cudahy[c]
[a]*CSIRO Mineral Resources, 26 Dick Perry Avenue, Kensington, WA 6151, Australia*
*(corresponding author: Carsten.Laukamp@csiro.au)*
[b]*MineSense Technologies, Vancouver, Canada*
[c]*C3DMM Pty Ltd, Perth, Australia*
**ABSTRACT**
The integration of surface and subsurface geoscience data is critical for efficient and
effective mineral exploration and mining. Publicly accessible datasets to evaluate the various
geoscience analytical tools and their effectiveness for characterisation of mineral assemblages
and lithologies or discrimination of ore from waste are however scarce. The open access
Rocklea Dome 3D Mineral Mapping Test Data Set (Laukamp, 2020;
https://doi.org/10.25919/5ed83bf55be6a) provides an opportunity for evaluating proximal and
remote sensing data, validated and calibrated by independent geochemical and mineralogical
analyses, for exploration of channel-iron deposits (CID) through cover. We present
hyperspectral airborne, surface and drill core reflectance spectra collected in the visible-near
infrared and shortwave infrared wavelength ranges (VNIR-SWIR; 350 to 2500 nm), as well as
whole rock geochemistry obtained by means of X-Ray fluorescence analysis and loss on
ignition measurements of drill core samples.
The integration of surface with subsurface hyperspectral data collected in the frame of
previously published Rocklea Dome 3D Mineral Mapping case studies demonstrated that about
30% of exploration drill holes were sunk into barren ground and could have been of better use,
located elsewhere, if airborne hyperspectral imagery had been consulted for drill hole planning.
The remote mapping of transported Tertiary detritals (i.e. potential hosts of channel iron ore



resources) versus weathered in situ Archaean geology (i.e. barren ground) has significant
implications for other areas where "cover" (i.e. regolith and/or sediments covering bedrock
hosting mineral deposits) hinders mineral exploration. Hyperspectral remote sensing represents
a cost-effective method for regolith landform mapping required for planning drilling programs.
In the Rocklea Dome area, vegetation unmixing methods applied to airborne hyperspectral
data, integrated with subsurface data, resulted in seamless mapping of ore zones from the
weathered surface to the base of the CID – a concept that can be applied to other mineral
exploration and mineral deposit studies. Furthermore, the associated, independent calibration
data allowed to quantify iron oxide phases and associated mineralogy from hyperspectral data.
Using the Rocklea Dome data set, novel geostatistical clustering methods were applied to the
drill core data sets for ore body domaining that introduced scientific rigour to a traditionally
subjective procedure, resulting in reproducible objective domains that are critical for the
mining process.

Beyond the already published case studies, the Rocklea Dome 3D Mineral Mapping

Test Data Set has the potential to develop new methods for advanced resource characterisation
and develop new applications that aid exploration for mineral deposits through cover. The here
newly presented white mica and chlorite abundance maps derived from airborne hyperspectral
highlight the additional applications of remote sensing for geological mapping and could help
to evaluate newly launched hyper- and multispectral spaceborne systems for geoscience and
mineral exploration.

**Key words**: Channel Iron ore Deposits, regolith, hyperspectral remote sensing, hyperspectral
drill core sensing, geochemistry



## 1. INTRODUCTION



The three dimensional (3D) geologic case history of the Rocklea Dome located in the
Hamersley Province (Western Australia) targeted the use of reflectance and emission
spectroscopy for measuring mineralogy and geochemistry specific to the exploration and
characterisation of economic Tertiary channel iron ore deposits in a terrain obscured by
weathered, transported materials. This public case history was generated by CSIRO's Western
Australian Centre of Excellence for 3D Mineral Mapping (C3DMM), which was operated from
2009 to 2012 and had the primary aim of generating and demonstrating the capabilities for
"scalable" 3D mineral mapping from the continental to the prospect scales (Cudahy, 2016).
The Rocklea Dome project was established in collaboration with Murchison Metal Ltd, who
granted C3DMM access to their drill hole dataset, consisting of 14 diamond cores and 180
reverse circulation drill holes. These drill holes were designed using traditional exploration
mapping technologies, such as published geology maps and geophysical data (magnetics and
radiometrics).
Key achievements of the Rocklea Dome 3D Mineral Mapping case study include:
- Based on the kaolin crystallinity index derived from surface and sub-surface
hyperspectral data (Cudahy, 2016), drill holes were identified that were sunk at surface
into barren (i.e. bedrock) weathered material. That is, approximately one third of the drill
holes need not have been drilled or would have been located differently had surface
mineral mapping data, e.g. airborne hyperspectral imagery, been used during drill hole
planning. This represents potential significant savings in time, money and environmental
disturbance.
- Characterisation of clay mineralogy associated with distinct domains of the CID and its
cover (i.e. kaolin group vs. Al-smectites vs. Fe-smectites) suggested that clay mineral



assemblages as well as calcrete atop buried CIDs have a different composition when
compared to regolith covering adjacent areas. That could represent useful information
when exploring for CIDs through regolith cover.
• Quantification of iron oxide phases and associated mineralogy derived from
hyperspectral data and validated using X-ray diffractometry and geochemistry (Haest et
al., 2012 a,b):
o iron (oxyhydr-)oxide content: RMSE of 9.1 weight % Fe
o Al clay content: RMSE 3.9 weight % $Al_2O_3$
o hematite/goethite ratio: RMSE 9.0 weight % goethite
o spatial characterisation of vitreous vs. ochreous goethite
• Geological modelling the iron ore resource of the Rocklea Dome CID (Haest et al., 2012
a,b; Cudahy, 2016; Fouedjio et al., 2018), which was reported by Dragon Resources in
2012 to be 72.6Mt (at 53% Fe cut-off) with 54.4% Fe, 7.2% $SiO_2$, 2.7% $Al_2O_3$, 0.031%
P and 11.2% LOI.
• Improvement of quality of mineral maps by application of vegetation unmixing methods
(Haest et al, 2013), resulting in seamless mapping of ore zones from the weathered
surface to the base of deposit (Cudahy, 2016).
All the above points showcase how hyperspectral data can be used for critical parts of
the mining cycle, especially exploration and 3D resource characterisation.
This article aims to provide an overview of the publicly available hyperspectral data set
of the Rocklea Dome, which ought to be used as a test data set for 1) data mining for exploration
and mining, 2) integration of independent geoscience data sets (i.e. hyperspectral,
geochemical), 3) resource modelling, and 4) different approaches for routine processing of
hyperspectral data.





The geological setting of the Rocklea Dome area, as well as analytical and processing

methods will be discussed first, after which the publicly available test data are listed as a table.
Example applications of the geochemical and mineralogical data for exploration, 3D mineral
mapping and Resource estimation are summarised briefly in the discussion.

**2. GEOLOGICAL SETTING**

The Rocklea Dome Channel Iron Deposit is located in the Hamersley Province, which

is the dominant source of Australia's iron ore exports. Channel Iron Deposits (CID) are
economically significant formations, providing a substantial percentage of the iron ore mined
in Australia. A detailed overview of the geology of the Rocklea Dome and the formation of the
CID was provided by Haest et al. (2012b) and is briefly summarised here. The bedrock geology
of the Rocklea Dome comprises a monzogranite pluton and cross-cutting mafic and ultramafic
intrusives that form part of the Pilbara Craton. The Archean age pluton is overlain by Archaean
to Proterozoic metasedimentary and volcanic rocks of the Hamersley Basin, enveloping the
central monzogranite dome (Fig. 1; Thorne & Tyler, 1996). Folding is attributed to
development of both the Ophthalmia and the Ashburton Fold Belt (Thorne & Tyler, 1996).

A meandering Tertiary palaeochannel passes over the Archean and Proterozoic rocks,

containing locally Channel Iron ore Deposits (CID), such as the Beasley River CID, which
crosscuts the north-western part of the Rocklea Dome. Channel iron ore was also drilled along
8 km strike length of a palaeochannel on the eastern side of the Rocklea Dome, which was
described by Haest et al. (2012a,b; 2013) as the Rocklea Dome CID (Figure 1). The bedrocks
and Tertiary channel are covered partly by regolith (e.g. Quaternary alluvium). Green
vegetation and dry vegetation (mostly Spinifex grass and bushes) cover the area partly.



A mixture of Fe-Ox pelletoids and ferruginised wood fragments below 10 mm in size
represent the major components of CIDs (Morris & Ramanaidou, 2007). In CID systems, the
base of the channel often consists of a clay horizon of variable composition. The CID is capped
in places by calcrete and silcrete.

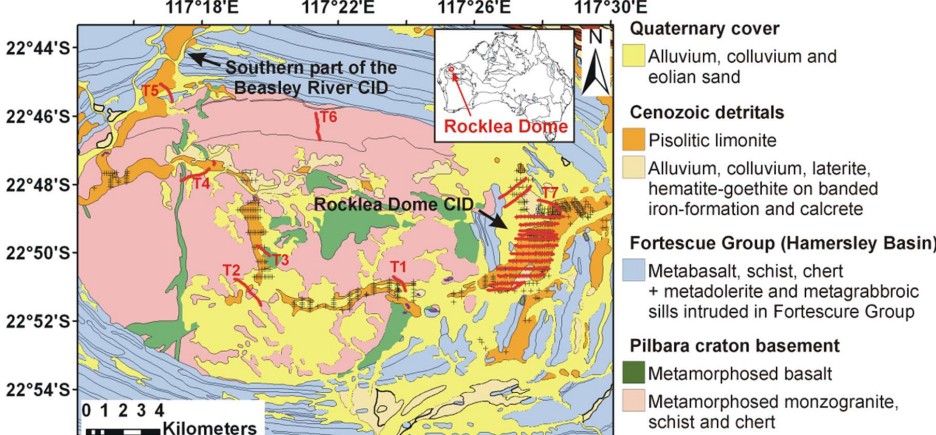

Figure 1: Geological map of the core of the Rocklea Dome (Haest et al., 2013; Thorne and
Tyler, 1996). Validation transects are indicated as red lines. T1 to T7 refer to transects
described in Haest et al. (2012a). The black crosses identify the position of all reverse
circulation drill cores intersecting the palaeochannel (i.e. pisolitic limonite in map) in the core
of the Rocklea Dome.





## 3. METHODS AND MATERIALS

### 3.1. *VNIR-SWIR drill core spectroscopy*

Reflectance spectra of 180 rock chips (RC) and 14 diamond drill cores (RKD) were measured using CSIRO's HyChips™ system (cf. Huntington et al., 2004), which comprises a TerraSpec™-based spectrometer (Malvern-Panalytical) system. In total, 7,520 reflectance spectra were collected from RC samples and 66,853 reflectance spectra were collected from RKD samples (Haest et al., 2012a). An automated X-Y table moves the drill core tray in a snake-like pattern below the TerraSpec optical fibre at a distance of ~6 to 13 cm (depending on sample type, i.e., diamond core or drill chips), while the spectral data are collected. Each sample spectrum is collected from a $1 \times 1$ cm area. Four light globes are positioned 40 cm above and at a small angle (off the backscatter/specular angle) to the measurement/sample point. In addition to hyperspectral data, high spatial resolution (0.1 mm pixel) images are collected from the core or chip tray and the sample height in the tray is measured using a laser profilometer. Reflectance spectra were calibrated using a Teflon/Spectralon™ panel (see Haest et al., 2012a for more details). TerraSpec™ spectra were collected in the visible to near-infrared (VNIR: 380−1,000 nm) and shortwave-infrared (SWIR: 1,000−2,500 nm), with sampling intervals of 1.4 nm in the VNIR and 2.0 nm in the SWIR and a wavelength accuracy of ±1 nm. The spectral resolution is 5 nm in the VNIR and between 11 and 12 nm in the SWIR. The TerraSpec™ radiance spectra of each sample are first converted to apparent bidirectional reflectance using the Teflon signal, which is collected at the beginning/end of each drill core/drill chip tray measurement cycle. This signal is then converted to absolute reflectance, based on the measurement of a Spectralon panel.

### 3.2. *Remote Sensing*



Airborne VNIR–SWIR imagery was collected using the Airborne Multispectral

Scanner (AMS), which is an earlier version of HyVista Corporation's HyMap™ system (Cocks
et al., 1998). The AMS system collects 96 bands over the VNIR-SWIR, excluding the
atmospheric bands from ~1000 nm to ~1400 nm and from ~1800 to ~1950 nm, respectively.
For each spectral band, the average spacing of collected bands is 15 nm and the average full
width at half maximum is 17 nm. The AMS data over Rocklea Dome were collected in a north–
south direction between the 31st of July 2000 and the 2nd of August 2000, comprising a set of
14 flight lines, totalling in a combined length of ~280 km at a pixel size of approximately 7 m.
Atmospheric correction was done using MODTRAN5 (Berk et al., 2004, 2006) and SODA
(Rodger, 2011), based on a combination of the AMS at-sensor radiance with in-scene flight
parameters (e.g. latitude, longitude, sensor height, etc.). For more details about georeferencing
and mosaicking the single flight lines, see Haest et al. (2013).

*3.3.    X-ray fluorescence analysis (XRF)*

XRF analysis of 11,900 RC samples (1m interval) for weight percentages of Fe, P, S,

$SiO_2$, $Al_2O_3$, Mn, CaO, $K_2O$, MgO, and $TiO_2$ was conducted by Kalassay Ltd. (now Bureau
Veritas Minerals Pty Ltd., Western Australia). A Bruker Pioneer X-ray fluorescence instrument
with an end window 4 kW rhodium X-ray tube was used. Sample preparation included drying
at 105°C for 12 hr or for 1 hr, depending on whether the sample was wet or dry, respectively.
Samples were then crushed to a nominal 90% passing 75 μm. The sample powders were fused
in a Herzog automated (RF energized) fusion furnace and cast into 40 mm diameter beads using
a 12:22 flux containing 5% sodium nitrate. Matrix corrections were applied using a calculated
alpha correction for this combination of flux, tube, and instrument geometry. Previously
determined weight ranges were used for both the sample and the flux weight. Kalassay Ltd.
used lab duplicates, internationally certified reference materials, and reference materials of the





same ore type as standards and reported a precision better than 0.01% for all analyses. In order
to evaluate the accuracy of XRF analyses undertaken by Kalassay Ltd., duplicate samples were
also sent to the Amdel laboratory in Cannington (Western Australia), with good correlation
observed (Haest et al., 2012a).

*3.4.   Loss on ignition (LOI)*

In order to characterise the mineral assemblages present in the samples in more detail,

loss on ignition (i.e., LOI) measurements were undertaken on 11,900 RC samples to record the
mass loss of samples on heating (Haest et al., 2012a). A pre-dried portion of all samples was
heated in an electric furnace to 1,000°C. During this process, goethite will release its strongly
bonded water and its OH groups between 260° and 425°C (Strezov et al., 2010), organic matter
will be completely ignited by 550°C (Dean, 1974), aluminosilicate clay materials will
decompose between 530° and 605°C (Strezov et al., 2010), and inorganic carbon will be
oxidized and lost as $CO_2$ between 700° and 850°C (Dean, 1974).

*3.5.   Sample storage*

Drill core trays, field samples and XRF standards are all stored at the Australian

Resources Research Centre (ARRC) in Kensington (Western Australia). Samples can be
viewed and investigated at the ARRC, using local analytical facilities.

**4. SOFTWARE AND PROCESSING METHODS**
*4.1.   Processing of hyperspectral drill core data*

Hyperspectral drill core data were analysed using the CSIRO's The Spectral Geologist

software (TSG$^{TM}$) by interpreting the abundance, composition and/or crystallinity of selected





mineral groups and species using the Multiple Feature Extraction Method. A list of scripts
applied to the hyperspectral drill core and rock chip data can be found in Table 1.





Table 1: Base scripts and multiple feature extraction method scripts used for the Rocklea Dome
3D Mineral Mapping project.

| Product name | Minerals detected | Base algorithm | Filters/Masks | Lower stretch limit | Upper stretch limit (only applicable for composition products) | related publication |
|---|---|---|---|---|---|---|
| Ferric oxide abundance (Ferric_oxide_abundance.txt) | Hematite, goethite, jarosite, "limonite" | Continuum removed depth of the 900 nm absorption calculated using a fitted 2nd order polynomial between 776 and 1050 nm *900/D* | R450 > R1650 | 0.04: low content | | further developed on the basis of Haest et al. (2012a,b), which used a 4th order polynomial or 4 band ratio approach |
| Hematite-goethite distribution (Hematite-goethite_distr.txt) | Hematite-goethite ratio (Cudahy and Ramanaidou, 1997) | Continuum removed wavelength of the 900 nm absorption calculated using a fitted 2nd order polynomial between 776 and 1050 nm. 900W | R450 > R1650 + 900D >0.025 | ~890nm: more hematitic | ~910 nm: more goethitic | further developed on the basis of Haest et al. (2012a,b), which used a 4th order polynomial or 4 band ratio approach |
| Ferrous iron abundance (Ferrous iron abundance.txt) | $Fe^{2+}$ in silicates & carbonates, (Fe-chlorites, Fe-amphibole, Fe-pyroxene, Fe-olivine, Fe-carbonate) | $(R_{620}+R_{1650})/(R_{1047}+R_{1235})$ *Ferrous* | | ~1.005: low content | | Laukamp et al. (2012) |
| opaques2 (opaques2inv.txt) | "Reduced" materials such as carbon black, sulphides and magnetite as well as Mn oxides. | albedo @ 1650 nm *1650* | OPAQUES_450D1650 >0.25; albedo @ 1650 nm 1650 <30% | 2: low content | | |
| White mica and Al-smectite abundance (wmAlsmai.txt) | Abundance of white micas (e.g. illite, muscovite, paragonite, brammalite, phengite, lepidolite, margarite) and Al-smecties (montmorillonite, beidellite) | Relative absorption depth of the 2200 nm absorption for which the continuum is removed between 2120 and 2245, determined using a 3 band polynomial fit around the band with the lowest reflectance. *2200D* | $((R_{2138}+R_{2190})/(R_{2156}+R_{2179}) 2160D2190$ <1.063 | 0.02: low content | | further developed on the basis of Sonntag et al. (2012), which used a 4th order polynomial or 4 band ratio approach |
| White mica and Al-smectite composition (wmAlsmci.txt) | Tschermak substitution of white micas, ranging from paragonite, brammalite, to illite, muscovite to phengite, and Al-smecties, ranging from beidellite to montmorillonite. | Minimum wavelength of the 2200 nm absorption for which the continuum is removed between 2120 and 2245, determined using a 3 band polynomial fit around the band with the lowest reflectance. *2200W* | $((R_{2138}+R_{2190})/(R_{2156}+R_{2179}) 2160D2190$ <1.063 | 2180 nm: Al-rich mica (muscovite, illite, paragonite, brammalite, lepidolite) | 2220 nm: Al-poor mica (~phengite) | further developed on the basis of Sonntag et al. (2012), which used a 4th order polynomial or 4 band ratio approach |
| Kaolin abundance index | Kaolin group minerals, namely kaolinite halloysite, dickite and nacrite | 2200D (Normalized depth of a fitted 4th order polynomial between 2120 and 2245 nm) | 2160D $((R2138+R2190)/(R2156+R2179))>1.005$ | 0.02: low content | | Sonntag et al. (2012), Haest et al. (2012a,b) |
| Kaolin composition index | Composition and crystallinity of kaolin group minerals ranging from well-ordered kaolinite to halloysite to dickite (and nacrite) | $[(R2138+R2173)/(R2156)]/[(R2156+R2190)/R2173]$ | 2200D>0.005 | low values = low crystallinity | high values = high crystallinity | Sonntag et al. (2012), Haest et al. (2012a,b) |
| Carbonates abundance (carbi3pfit.txt) | carbonates vs. MgOH-bearing silicates, based on left-asymmetry of CO3 feature @ 2340 | Relative absorption depth of the 2340 nm absorption for which the continuum is removed between 2270 and 2370, determined using a 3 band polynomial fit around the band with the lowest reflectance. 2340D | 2340D>0.04, 2295 nm<2340W<2360nm, 2250D < 0.025, 2380D>0.1117*2340D+0.0002. Asymmetry of the 2340 absorption using a fitted 4th order polynomial between 2120 and 2370: 2340_left_asym > 1.13 | 0.05: low content | | further developed on the basis of Sonntag et al. (2012), which used a 4th order polynomial or 4 band ratio approach |
| Carbonate composition (carbci3pfit.txt) | separating calcite, dolomite, siderite, … | Minimum wavelength of the 2340 nm absorption for which the continuum is removed between 2270 and 2370, determined using a 3 band polynomial fit around the band with the lowest reflectance. 2340W | 2340D>0.04, 2295 nm<2340W<2360nm, 2250D < 0.025, 2380D>0.1117*2340D+0.0002. Asymmetry of the 2340 absorption using a fitted 4th order polynomial between 2120 and 2370: 2340_left_asym > 1.13 | 2303 nm: magnesite; 2326 nm: dolomite | 2343 nm: calcite | further developed on the basis of Sonntag et al. (2012), which used a 4th order polynomial or 4 band ratio approach |
| White mica (+Al-smectite) abundance, refined for airborne hyperspectral imagery | Abundance of white micas (e.g. illite, muscovite, paragonite, brammalite, phengite, lepidolite, margarite) and Al-smecties (montmorillonite, beidellite) | Relative absorption depth of the 2200 nm absorption for which the continuum is removed between 2120 and 2245, determined using a 3 band polynomial fit around the band with the lowest reflectance. *2200D* | $((R_{2138}+R_{2190})/(R_{2156}+R_{2179}) 2160D2190 <1$, 2200D2320D > 1.5 | 0.04: low content | 0.255: high content | further developed on the basis of Sonntag et al. (2012), which used a 4th order polynomial or 4 band ratio approach |
| Chlorite (+epidote, +biotite) abundance, refined for airborne hyperspectral imagery | Abundance of chlorite (e.g. clinochlore, chamosite), as well as members of the epidote and biotite mineral groups | $(R2227+R2275)/(R2241+R2259)$, *2250D* | 2250D > 1.01, & 2300 < 2320W < 2342 & 2240 < 2250W < 2260 | 1.01: low content | 1.04: high content | further developed on the basis of Sonntag et al. (2012) |





### 4.2. Image Processing


The processing strategy for generating geoscience products from AMS data, such as the
Kaolin Crystallinity (Table 1) builds on the quality control of the acquired data (Cudahy et al.,
2008). Well calibrated radiance-at-sensor or surface reflectance data are required for the
processing of airborne hyperspectral imagery. Commonly applied levelling and statistics-based
methods were avoided as these introduce undesirable scene-dependencies, making a
comparison of image products from different areas impossible. Physics-based reduction
models were applied to the remote sensing data, using the image processing software ENVI$^{TM}$.
Complicating effects were removed in their order of development (i.e. 1. instrument, 2.
atmospheric, 3. surface effects) through either normalization or offsets.

### 4.3. The Multiple Feature Extraction Method

In hyperspectral proximal (e.g. HyLogging$^{TM}$) and remote (e.g. AMS) sensing
technologies, the VNIR, SWIR, and thermal infrared (TIR: ca. 6000 – 14500 nm) wavelength
ranges are used to infer abundance and composition of various geological materials. The
relative intensity and wavelength position of absorption features in the reflectance spectra
relate to the physicochemical characteristics of the various minerals. Feature extraction
methods can be used to determine the mineralogy of a sample material (Cudahy et al., 2008).
The advantage the multiple feature extraction method (MFEM) is that the associated scripts
are not biased on a training dataset or spectral reference libraries, but are based only on the
visible and/or infrared active functional groups of minerals (see Laukamp et al., 2011, for more
details). As they are instrument independent, the same scripts can be applied to remote sensing
and proximal hyperspectral data, easing the integration of, for example, surface (e.g. HyMap)
and subsurface data (e.g. HyLogging$^{TM}$) for the purpose of visualisation in 3D or advanced

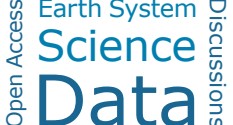

data analytics. Interferences of mineralogical information with other surface materials such as
vegetation can be evaluated by using a multiple linear regression model for unmixing
vegetation from hyperspectral remote sensing data (Rodger & Cudahy, 2009; Haest et al.,
2013). Other complications, such as spectrally overlapping materials, are removed by the
application of thresholds.

**5. DATA PRODUCTS AND APPLICATIONS**
Publicly accessible data of the Rocklea Dome 3D Mineral Mapping project can be
found          on          CSIRO's          Data          Access          Portal:
https://data.csiro.au/collections/#collection/CIcsiro:44783
(https://doi.org/10.25919/5ed83bf55be6a) and are listed in Table 2. Data and other content on
this site are scientific research data collected by CSIRO and third parties and are made available
on an 'as is' basis. If any data or other material are downloaded from this site, the user does so
at own risk and acknowledges that such data or other content: 1) may contain general
statements based on scientific research and may be incomplete and not applicable to all
situations; 2) is not professional, scientific, medical, technical or expert advice and is subject
to the usual uncertainties of scientific and technical research; and 3) should not be relied upon
as specific to you and therefore as the basis for doing or failing to do something. Expert
professional scientific and technical advice should be sought prior to acting in reliance on data
and other material from this site. To the extent permitted by law, CSIRO excludes all liability
to any person for any consequences, including but not limited to all losses, damages, costs,
expenses and any other compensation, arising directly or indirectly from using and any
information or material contained in it.



Table 2: Publicly accessible data of the Rocklea Dome 3D Mineral Mapping project
([https://data.csiro.au/collections/#collection/CIcsiro:44783](https://data.csiro.au/collections/#collection/CIcsiro:44783);
[https://doi.org/10.25919/5ed83bf55be6a](https://doi.org/10.25919/5ed83bf55be6a))

| main directory | sub directory | file name | type of data | source/IP |
|---|---|---|---|---|
| DTM | | DTM_Rocklea_50k.00t | Digital terrain model | GSWA |
| | | DTM_Rocklea_50k.dxf | | |
| | | DTM_Rocklea_50k.evf | | |
| | | DTM_Rocklea_50k.zip | | |
| | | Hardey_HR_DTM.00t | Digital terrain model of 100K mapsheet Hardey 2252 | GSWA |
| | | Hardey_HR_DTM.dxf | | |
| | | Hardey_HR_DTM.evf | | |
| | | Topography_ENVI | Digital elevation model | GSWA |
| | | Topography_ENVI.hdr | | |
| | | dem_plus_collars.csv | Digital elevation model | GSWA |
| drill hole data | RC_hyperspectral_geochem | GeoscienceProductDescriptions_ProximalHyperspectral.xlsx | table describing multiple feature extraction scripts applied to hyperspectral data for interpretation of mineralogy | CSIRO |
| | | RC_data.tsg | TSG-file | CSIRO |
| | | RC_data.ini | TSG-file | CSIRO |
| | | RC_data.bip | TSG-file | CSIRO |
| | | RC_data_cras.bip | TSG-file | CSIRO |
| | | RC_data_tsgexport.CSV | spectral and geochemical data exported from TSG | CSIRO |
| | RKD | RKD5-7-9.tsg | TSG-file | CSIRO |
| | | RKD5-7-9.ini | TSG-file | CSIRO |
| | | RKD5-7-9.bip | TSG-file | CSIRO |
| | | RKD5-7-9_cras.bip | TSG-file | CSIRO |
| remote sensing data | GeoTIFF_AMS/ | 2200D_Mstd.tfw | AMS product "2200D", showing the relative abundance of Al-clays | CSIRO |
| | | 2200D_Mstd.tif | | CSIRO |
| | | 2200WAR_2190-2205.tfw | AMS product "2200W", indicating compositional changes of Al-smectites and white micas $(Al^{VI}Al^{IV}(Fe,Mg)_{-1}Si_{-1})$ | CSIRO |
| | | 2200WAR_2190-2205.tif | | CSIRO |
| | | 2250_MStd.tfw | AMS product "2250D", showing the relative abundance of chlorite, epidote and/or biotite | CSIRO |
| | | 2250_MStd.tif | | CSIRO |
| | | 2330_2250-2380.tfw | AMS product "Carbonate abundance", showing the relative abundance of carbonates | CSIRO |
| | | 2330_2250-2380.tif | | CSIRO |
| | | KC_NoSM_22D+216DM_3MeFi.tfw | AMS product "Kaolin crystallinity" | CSIRO |
| | | KC_NoSM_22D+216DM_3MeFi.tif | | CSIRO |
| | TXT_AMS/ | 2320D_vegunm.txt | AMS product "2320D", vegetation unmixed | CSIRO |
| | | AlOHAbVegunm.txt | AMS product "Al-clay abundance index", vegetation unmixed | CSIRO |
| | | FeOxVegUnm.txt | AMS product "Ferric Oxide Abundance Index", vegetation unmixed | CSIRO |
| | | SRTM_RockleaDome+HardeyRiver.txt | Digital elevation model | GSWA |



| | StudentExercises_Rocklea.docx | Exercises for analysis of HyLogging data | CSIRO |
|---|---|---|---|
| Rocklea Dome exercise | Answers_CIDexercises.docx | Suggested answers to exercises for analysis of HyLogging data | CSIRO |
| | MinSpec_Workshop_7RockleaDomeTSG_HandsOn.pptx | PPT-presentation summarising Rocklea Dome exercise and results | CSIRO |


The following chapters briefly describe examples of how the provided hyperspectral
and geochemical proximal and remote sensing data sets can be used to address challenges for
the mineral resources sector.

*5.1.   Drill core mineralogy and geochemistry*
Reflectance spectra collected from rock chips (RC) and diamond drill cores (RKD)
using CSIRO's HyChips™ system presented a cost-effective way to spatially map the major
ore (i.e. goethite +/- hematite) and gangue minerals (i.e. kaolinite, smectite, carbonate), apart
from quartz, in detail. To achieve this, the relative intensity of mineral-diagnostic absorption
features was calculated using a suite of batch scripts ("Geoscience Products" in Haest et al.,
2012a). The relative intensity of the respective absorption features correlates with the relative
abundance of the respective mineral, whereas the wavelength position of key absorption
features relates to mineral speciation (e.g. ochreous versus vitreous goethite) or determining
the mineral chemistry. For example, the relative abundance of iron oxides was calculated from
the relative depth of the ferric iron-related absorption at around 900 nm (Cudahy &
Ramanaidou, 1997), whereas goethite was distinguished from hematite by tracking the
wavelength position of the same absorption feature (Table 2 in Haest et al., 2012a).
Whole rock geochemistry obtained from the same drill core material showed significant
correlations with the Geoscience Products. Haest et al. (2012) determined an RMSE of 9.1
weight % Fe for the correlation between the hyperspectrally-derived iron oxide abundance and
the XRF weight % Fe data and an RMSE of 3.9 weight % $Al_2O_3$ for correlation between the



hyperspectrally-derived Al-clay abundance and the XRF weight % $Al_2O_3$ data. The errors
associated with the correlations were found to be due to a combination of grain size variations
and the transopaque behaviour of iron oxides and/or different amounts of silica, causing
variations in the optical depth of sample material.

*5.2.    Surface mineral mapping*

Airborne hyperspectral surveys provide spatially contiguous mineralogical information

of the Earth's surface at high spatial resolution (down to circa 1 m). The relative intensity of
mineral diagnostic absorption features and their wavelength positions can be used to infer the
relative abundance of the respective minerals and even variations of single mineral species in
terms of their cation composition, crystallinity and hydroxylation. The Rocklea Dome case
study data set was used by Haest et al. (2013) to demonstrate how quantitative mineral maps
can be produced by validation of airborne hyperspectral data against field data, including
reflectance spectra and XRF data collected from surface samples. The effect of both green and
dry vegetation cover was unmixed at the pixel-level using the Normalised Difference
Vegetation Index (NDVI; e.g. Tucker, 1979) and the continuum-removed depth of the
cellulose-lignin absorption centred at around 2100 nm, respectively. The resulting mineral
mapping products have a higher spatial continuity, as well as higher accuracy of, for example,
mineral abundance or composition values shown in single pixels. This proved to be especially
useful in areas with outcropping CID, which appeared to be sub-economic from the original
iron oxide abundance mineral maps but showed as potentially economic CID resources when
the vegetation cover was unmixed (Figure 2).

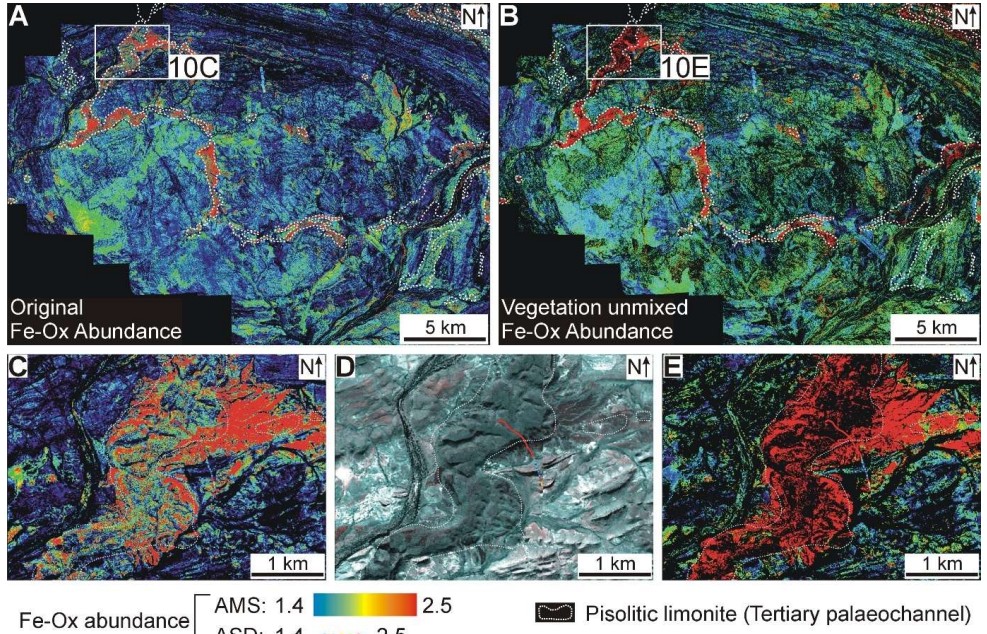

Figure 2: A–B: Fe-(oxyhydr-)oxide (Fe-Ox) abundance maps of the Rocklea Dome without
(A) and with (B) vegetation unmixing. C–E: Fe-Ox abundance maps of the southern part of
the Beasley River CID with (C) and without (E) vegetation removal and the false colour
image of this area (D). The Beasley River CID has a plateau like surface expression, with the
edges of the plateau clearly visible in the false colour image. These edges where mapped by
the Geological Survey of Western Australia as representing the boundary of the pisolitic
limonite (Fe-rich palaeochannel; white stippled line) (the Fe-Ox abundance measurements
collected along transects 1 to 7 with the TerraSpec[TM] are also shown for reference).

Beyond the iron oxide, kaolin and carbonate mineral maps published by Haest et al.
(2013), airborne hyperspectral data can be used to create numerous additional mineral mapping
products that can be used to address other geoscientific questions. For example, the Rocklea
Dome presents a wide variety of igneous units that are part of the Proterozoic basement of the
Pilbara Craton (Figure 1). These include 1) metamorphosed monzogranite, schist and chert, 2)
metamorphosed basalt, and 3) amphibolite dykes. According to the white mica abundance
derived from airborne hyperspectral data (green in Figure 3a), the metamorphosed
monzogranite contains less white mica, when compared to the metamorphosed schists which



are striking East-West and occur in the northern part of the Rocklea Dome (red in Figure 3a).
In the eastern half of the Proterozoic basement in the Rocklea Dome, white mica is much less
abundant to absent. This coincides with elevated amounts of chlorite (folded lithologies in the
centre of Figure 3b), which map out metamorphosed basalt (Figure 1). The North-South
striking occurrence of chlorite in the Western half of the Proterozoic basement traces an
amphibolite dyke. Both the white mica abundance and chlorite abundance maps can also be
used to map out different lithologies in the metasediments and metabasalts of the Fortescue
that crop out to the North and South of investigated area, demonstrating how the airborne
hyperspectral data can be used to map out all major lithologies occurring in the Rocklea Dome
case study area.

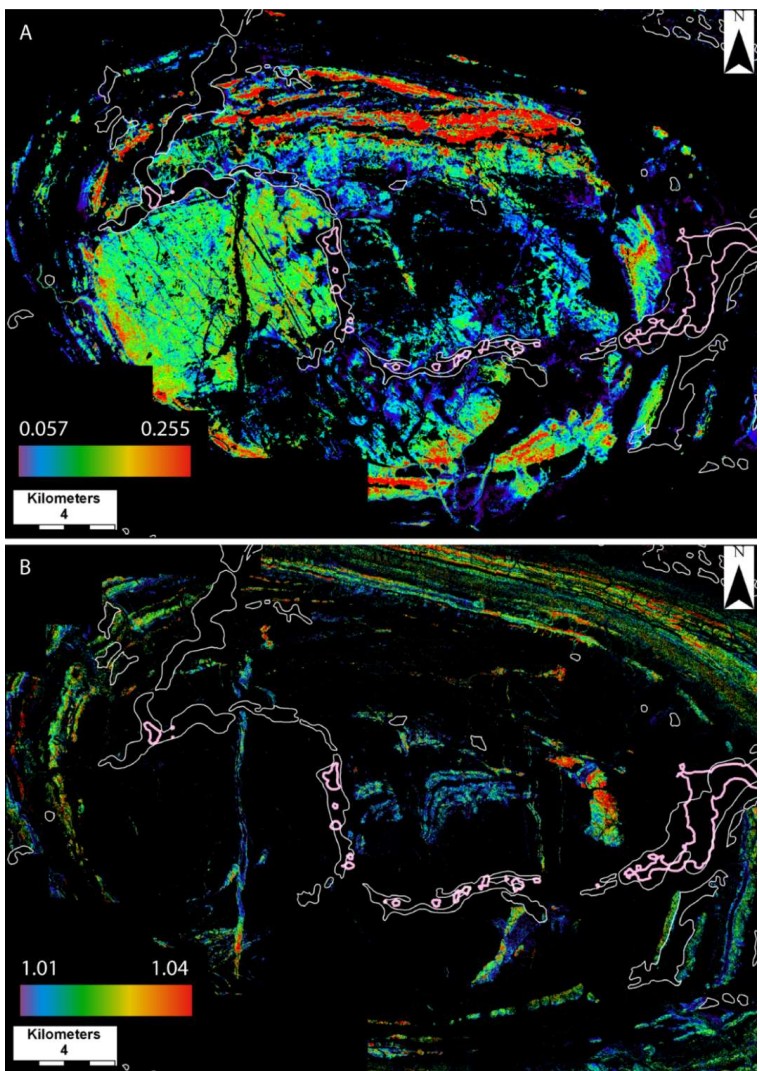


Figure 3: A & B: White mica (+Al-smectite) and chlorite (+epidote, +biotite) abundance
maps of the Rocklea Dome area in A and B, respectively, calculated from airborne
hyperspectral data using algorithms described in Table 1. Warm colours represent high
abundance and cool colours low abundance of the respective minerals. Black pixels have
been masked out as relative intensity of the absorption feature mapped in the respective
mineral map is below a given threshold (Table 1) and/or because of non-mineralogical effects
(e.g. vegetation, clouds). A shows monzogranites in the western part of the dome in green
colours and the Fortescue Group in the northern fringe of the dome in red colours. B
highlights Archean metamorphosed basalts in the eastern part of the dome structure and an N-
S trending amphibolite dyke in the western part of the dome. White lines indicate the surface
extension of the Tertiary channel as mapped by Thorne & Tyler (1996). Pink lines indicate
the horizontal extension of the Tertiary channel as mapped by the hyperspectral data.

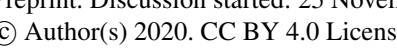


### 5.3.    *3D Mineral Mapping*

The hyperspectral drill core data can be combined with airborne hyperspectral data into

a seamless 3D mineral model of the Rocklea Dome CID (Figure 4). For this, all hyperspectral
data were resampled to the same spatial resolution and imported into the 3D modelling software
GoCad/SKUA$^{TM}$. The channel basement contact that was delineated at depth using the kaolin
crystallinity products could also be delineated at the surface from the airborne hyperspectral
image. A combination of both provided a seamless surface of the channel bottom (grey surface
in Figure 4) that separates the basement characterised by well-crystalline kaolinite from the
tertiary channel sediments characterised by poorly-crystalline kaolinite. The here identified
channel basement contact deviates at the surface significantly from the area mapped by the
geological survey as palaeochannel. This suggests that drilling patterns could have been much
better defined if the airborne hyperspectral-based surface outline would have been available
prior to drilling (Cudahy, 2016).

As part of their 3D Geomodel Series, the Geological Survey of Western Australia

(GSWA) provides access to 3D models of the Rocklea Dome area via their online portal:
https://dmpbookshop.eruditetechnologies.com.au/product/rocklea-inlier-2016-3d-geomodel-
series.do. The data can be viewed in three different formats (3D PDF, Geoscience Analyst,
GOCAD).

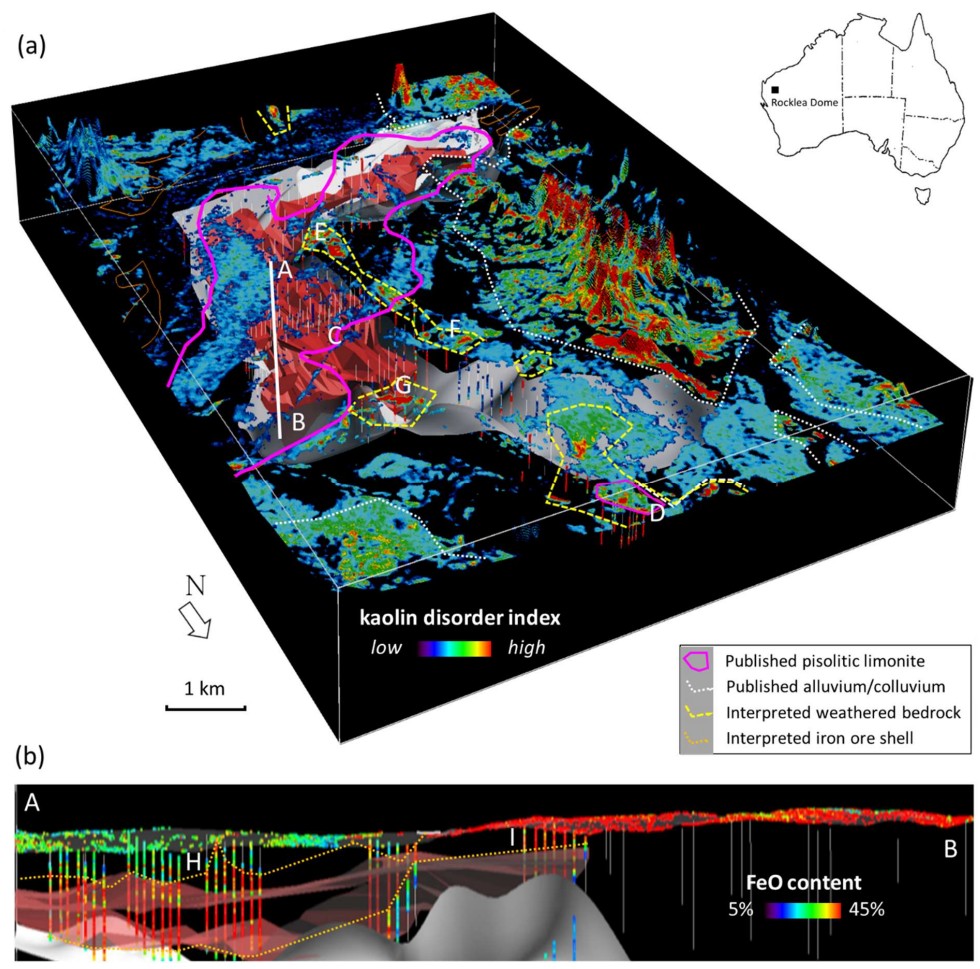

Figure 4: 3D mineral models of the Rocklea Dome area (Cudahy, 2016). Scene centre is approximately 22.8216° latitude 117.4652° longitude. (a) A southwest oblique 3D view of the Rocklea Dome study area showing kaolin disorder measured using airborne HyMap™ (surface) and drill core HyLogger™ (coloured vertical pegs) reflectance spectra. Warmer colours (well-ordered kaolin) relate to weathered, in situ bedrock, while cooler colours (poorly-ordered kaolin) relate to transported (alluvium/colluvium) materials. The interpolated model of the base of the channel iron system calculated using the 3D kaolin crystallinity map is shown by the shaded grey surface. The CID, which was calculated from the XRF-derived % FeO (Haest et al., 2012a), is shown by a shaded red volume. Areas of weathered bedrock (as mapped by Haest et al., 2012b and Cudahy, 2016) are highlighted by yellow-coloured hashed lines and highlight which drill cores were sunk into barren ground. A white straight line shows the location of the cross-section (A–B) presented in (b); (b) Cross section A–B in (a) of the % FeO measured from the drill core and airborne imagery, which was vegetation unmixed (Haest et al., 2013).

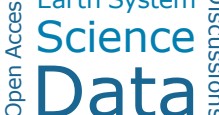


*5.4.    Resource estimation*

Resource estimation of base and precious metal deposits requires the grouping of drill
hole data into domains that represent zones of homogenous properties for accurate grade
estimation and practical exploitation purposes. In practice, this is more than often performed
through a subjective time-consuming manual interpretation of sample analytical data.
Traditional automated clustering techniques, such as multivariate clustering and k-means, tend
to show poor spatial contiguity of domains in a mineral deposit. Fouedjio et al. (2017) used the
Rocklea Dome drill core data set to showcase how geostatistical clustering methods can take
spatial dependency into account (Figure 5). By integrating whole rock geochemistry and
hyperspectral drill core data, Fouedjio et al. (2017) revealed two distinct domains in the
Rocklea Dome Channel Iron Ore Deposit that are mainly characterised by four geochemical
variables (FeO, $Al_2O_3$, $SiO_2$ and $TiO_2$) and two mineralogical variables derived from
hyperspectral data (ferric oxide abundance and kaolinite abundance). Ore body domaining
through geostatistical clustering represents a method for objective samples clustering that
introduces scientific rigour to a traditionally subjective procedure. The robust domaining is
based in genuine multivariate geostatistics combining all available data. The flexible and
reproducible automatic domaining technique saves time, improves the understanding of
domains critical for exploitation of the ore and allows an easy integration of new data sets.

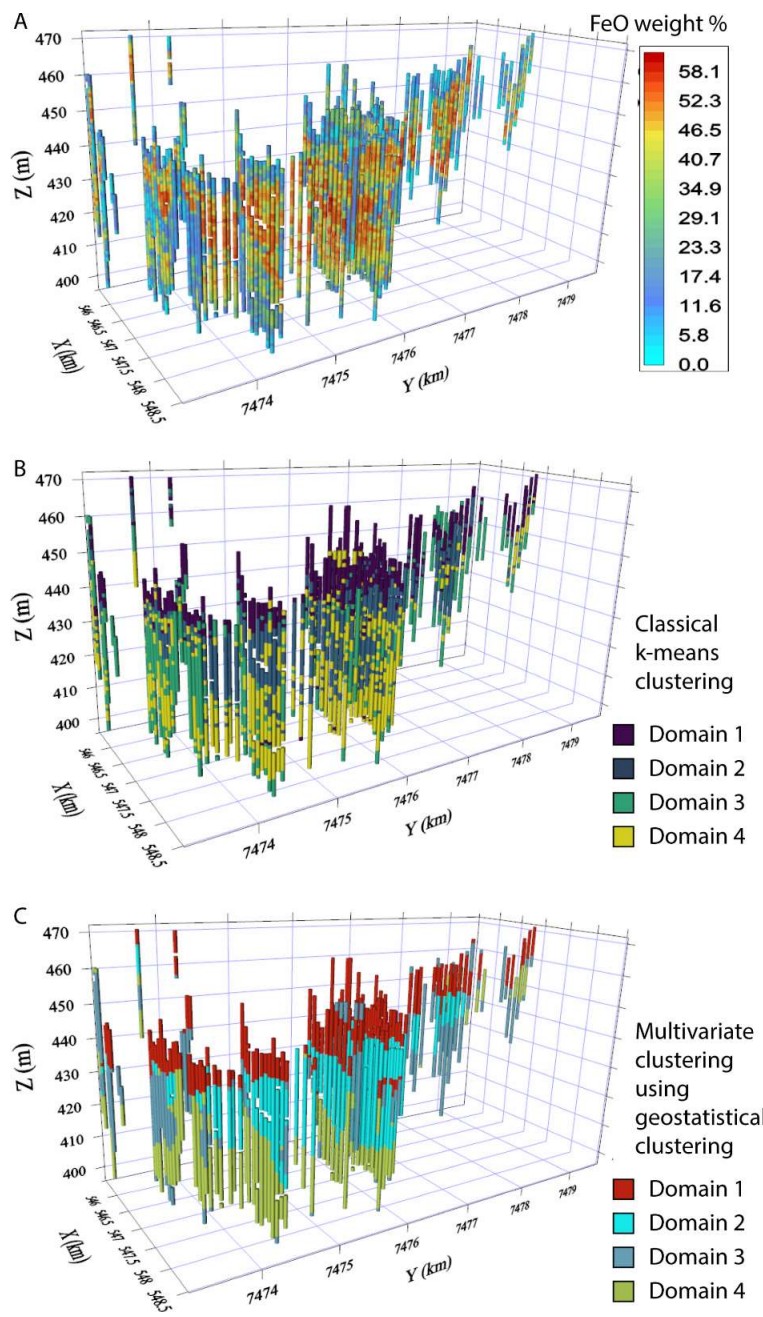


Figure 5: a) Spatial plot of FeO distribution in the Rocklea Dome Chanel Iron Deposit. (b)
Classical k-means clustering method using 4 domains. (c) Geostatistical spectral clustering
using 4 domains. Z-axis was scaled to ease visualisation. Modified from Fouedjio et al.
(2017).






*5.5.    Teaching material*

The publicly available Rocklea Dome data set provided an opportunity to compile

training and teaching material about the application of hyperspectral drill core and chips data
for iron ore resource characterisation using The Spectral Geologist Software (TSG$^{TM}$;
https://research.csiro.au/thespectralgeologist/). Student exercises and example answers, as well
as a ppt for teaching are part of the data package:

Exercise: StudentExercises_Rocklea.docx

Answers: Answers_CIDexercises.docx

PPT for teaching: MinSpec_Workshop_7RockleaDomeTSG_HandsOn.pptx


**6. CONCLUSIONS AND OUTLOOK**

We have established an open-access dataset comprising drill core, surface and airborne

hyperspectral data of the Rocklea Dome area in the Hamersley Basin of Western Australia,
which features a wide variety of lithologies and morphologies and is prospective for channel-
hosted iron ore resources. The proximal and remote sensing data, together with associated
whole rock geochemistry are ideal for researching the geology of this economically significant
area and allow a thorough comparison of different geoanalytical techniques and their
effectiveness for resource characterisation. Combining the surface and subsurface data into 3D
mineral maps provides a better visual understanding of the geological environment.

In addition to the already published surface and subsurface mineral mapping products,

many more Geoscience Products can be generated to better understand this geologically
complex area. The here newly presented white mica and chlorite abundance maps clearly



highlight the potential for mapping out different sections of the Archaean monzogranitic
basement as well as different generations of mafic intrusives. Of particular interest are the
contact zones between the mafic dykes and their host rocks, as they could help to better
understand the intensity of alteration within the dyke and within the host granite as well as the
associated fluid-rock interaction processes.
The teaching material provided together with this open-access dataset aims to support
training of geoscience graduates and post-graduates in the potential applications of
hyperspectral proximal and remote sensing data for mineral exploration and resource
characterisation.
All analytical technologies used for collection of the geoscience data, as well as
software packages used for processing the data, are commercially available. However, it should
be noted that the HyChips™ system is now superseded by HyLogger3, which collects thermal
infrared wavelengths (TIR; 6000 to 14500 nm) in addition to the VNIR-SWIR data. The
collection of the TIR wavelength range enables the characterisation of major rock forming
minerals such as quartz, which are of major importance for characterisation of iron ore
resources, but were not detectable with HyChips$^{TM}$. The HyLogger3 technology is in operation
at    the    six    nodes    of    the    Australian    National    Virtual    Core    Library
(https://www.auscope.org.au/nvcl), which provides online open access to more than 3,500 drill
cores from the Australian continent.

**7. DATA AVAILABILITY**
The    supplement    related    to    this    article    is    available    online    at:
https://doi.org/10.25919/5ed83bf55be6a (Laukamp, 2020). A 3D model of the Rocklea Dome
data set is also available from the Geological Survey of Western Australia:
https://dasc.dmp.wa.gov.au/DASC?productAlias=Rocklea3D.




## 8. AUTHOR CONTRIBUTIONS

CL, TC and MH contributed equally to the manuscript preparation. CL is the custodian
of the Rocklea Dome data set stored on the CSIRO's Data Access Portal.

## 9. COMPETING INTERESTS

The authors declare that they have no conflict of interest.

## 10. ACKNOWLEDGEMENTS

The Rocklea Dome 3D Mineral Mapping project was funded by the Western Australian
Government through their support to the Western Australian Centre of Excellence for Three-
dimensional Mineral Mapping (C3DMM) in Kensington and by Murchison Metals Ltd. M.
Cardy, A. Hackett, and S. Travaglione are acknowledged for the acquisition of the infrared
spectroscopic data. This work profited from fruitful discussions with CSIRO colleagues C.
Ong, A. Rodger, E. Ramanaidou, and M. Wells and with Murchison Metals Ltd. geologists J.
Johnson and S. Peterson. E. Ramanaidou (CSIRO) as well as staff of Murchison Metals were
instrumental in securing this site for public demonstration. The Geological Survey of Western
Australia covered part of the costs for the diamond drilling through their Exploration Incentive
Scheme co-funded Exploration Drilling program.

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
