# Peer review of "The Rocklea Dome 3D Mineral Mapping Test Data Set"

_Earth System Science Data, 2020_

## Referee Comment (RC1) · Anonymous Referee #1 · 5 Jan 2021

My field of expertise covers geostatistics, but not geology, so my review will not cover the geological aspects of the work.

The article describes publicly accessible datasets for characterisation of mineral assemblages and lithologies in Australia. The open access dataset is obtained by combining several data sources and provides a state-of-the-art dataset for geological modeling. Standard techniques are used for validation and verification. The presentation of the manuscript is clear and concise. The Introduction highlights the benefit of the study and specifically of the use of hyperspectral data.

The strong point of the study lies perhaps in the efforts made by the authors to present the datasets through very informative figures (e.g. Fig.4), which are properly described in the text. The combination of surface and subsurface observations into 3D maps is

well described.

In conclusion, the study is valuable. My advice to the editor is to publish it as it is.

---

## Referee Comment (RC2) · Anonymous Referee #2 · 21 Jan 2021

Manuscript: The Rocklea Dome 3D Mineral Mapping Test Data Set

Overview comments:

This research paper presents an open-access dataset that integrates multiple geochemical and mineralogical data collected from multiple proximal and remote sensing techniques. This publicly accessible dataset has carried out in the Rocklea Dome located in the Hamersley Province (Western Australia).

The work is well written and illustrated with figures and tables, and their conclusions seem to be consistent with the results of data products.

General vote: manuscript may become acceptable after minor changes, which are detailed in this review (see 'Line-by-Line commentary').

[Figure]

Line-by-Line commentary:

Line 20: 2,500 nm (add the thousand comma separator). Lines 27: Transported by fluvial streams? Please, specify. Lines 27, 56, 117, 122 and 361: Cenozoic instead of Tertiary (indicate its precise age if possible. For example, Miocene). Line 28: Change Archaean geology by Archaean-rock outcrops or Archaean bedrock. Lines 68 to 72: Consider to re-write; the text is confusing. Lines 117 to 123: Explain briefly the formation of CIDs. Line 126: Palaeochannel instead of channel. Line 164: ... ∼1,000 nm to ∼1,400 nm ...∼1,800 nm to ∼1,950 nm. Line 181: 'energized' and 'diameter' are examples of American English spelling. Please, use the same style of English spelling along the manuscript (American or British English spelling). In some cases, you have used British English spelling. Line 193: In the other parts of the text, authors do not use comma after 'e.g.' and 'i.e.'. Please, unify this style issue. Lines 195 to 199: Consider to use the Present Simple tense in this case. Line 196: 260°C and 425°C.The same occurs in lines 198 and 199. Line 213: Table 1: Add spaces (∼890 nm and ∼910 nm). Composition instead of 'composotion'. Polynomial instead of 'Polynomila'. In addition, in many cases, the measurement unit is absent as well as the thousand comma separator such as: Relative absorption depth of the 2,200 nm absorption for which the continuum is removed between 2,120 nm and 2,245 nm, determined using a 3 band polynomial fit around the band with the lowest reflectance. 2200D. Line 229: 6,000 − 14,500 nm Line 230: '...and composition of various geological materials.' I suggest to modify this sentence like follows: ...and mineralogical composition of rocks and sediments. Line 234: The advantage of the multiple feature extraction... Line 235: based on instead on 'baised on'. Lines 296: Consider to change 'down to circa 1 m' by '≤1 m' or '∼1 m'. Line 305: 2,100 nm. Lines 350, 351 and 428: Palaeochannel. Line 408: Channel instead of 'chanel'.

Figures:

Figures 1 and 3: Consider to change Kilometers by km. Figure 4: Please, indicate what represents the letters from C to I in Fig. 4.

Please also note the supplement to this comment:
https://essd.copernicus.org/preprints/essd-2020-336/essd-2020-336-RC2-
supplement.pdf

---

## Author Comment (AC1) · 25 Jan 2021

Thanks to Reviewer 2 for the constructive feedback. The text was revised accordingly. The responses below refer to items that have not been changed. Response to RC2 Lines 27, 56, 117, 122 and 361 < The age description was adopted from previously published papers about the Rocklea Dome area. As no additional work in terms of dating was done by our work, the authors prefer to adopt the published description. Response to RC2 117 to 123: Explain briefly the formation of CIDs <The description of the formation of CIDs would require quite a lot of text and is beyond the scope of this paper. As mentioned in the lines above line 117, a description of CID formation can be found in Haest et al. (2012b). Response to RC2 Line 230: 'and composition of various geological materials.' I suggest to modify this sentence like follows 'and mineralogical

composition of rocks and sediments' < Using "rocks and sediments" would restrict the group of material that can be analysed using HyLogger to a small subset of material that is analysed with these instruments in operation (e.g. Australian State and Territory Geological Surveys). A few words were added to the text to make that clear. Response to RC2 Lines 296: Consider to change 'down to circa 1 m' by'1 m' or '1 m'. < The original text was kept, as the authors wanted to make clear that the spatial resolution of remote sensing data sets, typically, can be as detailed as 1 m, but covers a wide range.
* * *

---

## Author Comment (AC2) · 25 Jan 2021

Thanks for reviewing the manuscript and many thanks for the kind feedback.